# Gratitude to God: A Unique Construct Adding to Our Understanding of Religiousness and Gratitude

**Crystal L. Park** [1,*], **Joshua A. Wilt** [2] and **Adam B. David** [1]

1. Department of Psychological Sciences, University of Connecticut, Storrs, CT 06269, USA
2. Department of Psychological Sciences, Case Western Reserve University, Cleveland, OH 44106, USA
* Correspondence: crystal.park@uconn.edu

**Abstract:** In two national samples in the United States, we aimed to determine the extent to which GTG is distinct from both general gratitude and general religiousness, using statistical methods to determine (1) if GTG shows patterns of association with other variables distinct from general gratitude and religiousness, and (2) whether GTG predicts wellbeing above and beyond both general gratitude and religiousness. Online studies were conducted with 267 (Study 1) and 184 (Study 2) adults. Results across the two studies were consistent in demonstrating that GTG shows associations with relevant constructs that are distinct from both general religiousness and general gratitude. Further, GTG independently predicted aspects of psychological wellbeing, although findings were not consistent across all aspects. These findings indicate GTG is a unique construct warranting future research.

**Keywords:** gratitude; religion; religiousness

## 1. Introduction

Building on the proliferating empirical work on gratitude (Jans-Beken et al. 2020; Portocarrero et al. 2020) and the longstanding theological traditions of gratitude to God embedded within most world religions (Tsang and Martin 2013), attention has recently turned towards empirical work on gratitude to God. Although gratitude is central to many of the world's religions, little psychological research has explored religious gratitude, or more specifically, gratitude to God (GTG). The current set of studies aimed to determine the extent to which GTG constitutes a unique construct distinct from the more general constructs of gratitude and religiousness.

GTG is clearly relevant and important to many people, and studies have shown that people tend to report high levels of GTG. For example, in a national survey of Americans, 52% reported feeling regularly grateful to God while 44% saw gratitude as an expression of love for God or a higher power (John Templeton Foundation Gratitude Survey 2012). Preliminary work on GTG has found it to be associated with salutary mental and physical health consequences (e.g., Krause and Hayward 2015; Krause et al. 2017).

However, foundational questions about GTG remain. In particular, as the field matures, determining whether GTG constitutes a distinct construct or instead simply represents a subtype of general religiousness or general gratitude is critical to building a robust science. Considering GTG in this way does not diminish its potentially deep significance at a theological or experiential level, both of which are well-established (e.g., Emmons and Crumpler 2000; Townes 2021). Clearly GTG is substantively different in many ways from gratitude as it is usually studied (Rosmarin et al. 2011) and is also conceptually not the same as general religiousness. However, to constitute a focus of empirical psychological research, the construct of GTG must demonstrate some "added value" regarding its explanation of human experience rather than being wholly redundant with already well-studied constructs. That is, we already know a great deal about gratitude and about religiousness, but it remains to be seen if GTG comprises a unique construct.

One way to determine the distinctiveness of GTG relative to more general gratitude or religiousness is to compare patterns of associations of these three constructs with other constructs that might be expected to be especially strongly associated with GTG, such as one's views of and relationship with God. This approach was taken in a study of Iranian Muslim university students that compared associations of general gratitude and religious gratitude with personality and wellbeing. This study found similar patterns between the two types of gratitude, but most associations appeared stronger for general gratitude (Aghababaei and Tabik 2013). Similar results were reported in a series of subsequent studies of Iranian and Polish university students (Aghababaei et al. 2018). The latter set of studies, however, also included measures of religiousness and found that GTG related more strongly to intrinsic and extrinsic religiousness than did general gratitude (in the Polish sample only), suggesting that GTG and general gratitude may be somewhat distinct. However, this study did not test the extent to which GTG was distinct from general religiousness.

Aside from general religiousness, other constructs may also relate differentially to gratitude to God. For example, preliminary but intriguing research suggests that suffering and hardship may promote transformative personal development that may manifest as GTG (Krause et al. 2012). Thus, those who have survived major adversity may generally be more grateful to God. For example, suffering serious health issues often leads individuals to experience deep feelings of gratitude and a deepened relationship with God (Chun and Lee 2013; Krause et al. 2012; Sacco et al. 2014). Our research to date on major life stressors and trauma has also found that religiousness is a robust predictor of people's perceptions that they have grown through hardship and become closer to God (Park 2006, 2013). This preliminary evidence suggests that GTG may be a common emotional experience in the wake of high-magnitude stressful events, particularly for religious people. Curiously, the influence of major positive life events has received relatively little attention in the general psychological literature compared to the impact of trauma and adversity (e.g., García-Bajos and Migueles 2013; Collins et al. 2007). It is reasonable to conjecture that individuals who have experienced more major positive life events would feel more gratitude to God, although no research to date has examined this notion.

Another factor that may distinguish among general religiousness, general gratitude, and GTG is views of and relationships with God. If GTG is distinct from general gratitude and general religiousness, we would expect that GTG would relate more strongly to some religious variables such as views of God as benevolent or finding comfort from God than would either general religiousness or general gratitude; conversely, views of an authoritarian or malevolent God or spiritual strain might demonstrate a weaker relationship with GTG than with general religiousness or gratitude.

A second, perhaps more sophisticated, way to determine the distinctiveness and value of the construct of GTG is to examine whether it predicts important "outcomes" such as wellbeing even when taking into account the influence of already-established constructs such as general religiousness or gratitude. In this way, GTG could be established as a unique construct. However, although GTG has been related to health and wellbeing in several studies (e.g., Krause et al. 2014), few of these studies took into account (i.e., included and statistically controlled for) general gratitude, which is a well-documented robust predictor of health and wellbeing (e.g., Jans-Beken et al. 2020).

When general gratitude is taken into consideration statistically, studies generally show diminished associations of GTG with hypothesized variables such as wellbeing. For example, one study showed that while GTG was related at a bivariate level with mental and physical health indices, these associations disappeared when controlling for general gratitude (Rosmarin et al. 2011). Similarly, the above-cited study with three samples (two of Iranian Muslims in Iran and one of Polish Christians) found very few independent effects of GTG on a host of variables once general gratitude was taken into account (Aghababaei et al. 2018). Thus, we aimed to determine whether GTG predicted wellbeing after taking general gratitude into account.

An additional concern, that GTG is simply a subcomponent of more general religiousness, also remains to be addressed. Given its prominent role in most monotheistic religious

teachings (Emmons and Crumpler 2000; Portocarrero et al. 2020), higher religiousness would be expected to be associated with higher GTG. However, whether GTG merely reflects this larger construct of general religiousness or whether specifically assessing one's GTG adds predictive power to models of the effects of religiousness on outcome variables remains to be determined. Some studies have included and controlled for some aspect of religiousness, but these have mostly been behavioral aspects such as service attendance rather than internal or experiential aspects. Even so, these studies tend to find minimal residual effects for GTG and typically only for subsets of the sample (e.g., Krause et al. 2017).

One proposed way in which GTG may be distinct and not simply a subset of general gratitude or religiousness is that GTG may be especially relevant to certain groups, thus exerting robust effects in these groups that go above and beyond general gratitude or religiousness. For example, for highly religious individuals, GTG may be highly valued and cultivated and its effects might extend beyond general gratitude. Indeed, in the above-cited study by Rosmarin et al. (2011) that controlled for general gratitude, GTG did add additional predictive power over and above regular gratitude, *but only for the subset of the sample high in religiousness.*

In the current set of two studies, we aimed to determine the extent to which GTG is distinct from both general gratitude and general religiousness, using statistical methods to determine (1) if GTG shows patterns of association with other variables distinct from general gratitude and religiousness, and (2) whether GTG predicts wellbeing above and beyond both general gratitude and religiousness overall and/or in the subset of people who are particularly religious.

## 2. Study 1

In Study 1, we first aimed to determine if GTG related especially strongly to a variety of constructs that might be more relevant to it than to general gratitude or general religiousness. One such construct is life event history, particularly major positive and negative events. We reasoned that people who have experienced more good fortune (i.e., more positive life events) and less adversity (i.e., fewer negative life events) would feel more gratitude, especially to God, although we were unable to find any previous research that has tested this proposition. We also included experiential aspects of religiousness, such as images of God and finding comfort from or feeling anger towards God. We anticipated that finding comfort from God and having benign views of God would be particularly strongly associated with GTG while spiritual strain and negative God images would be particularly inversely strongly associated with GTG. Our second aim was to determine whether GTG was associated with wellbeing after taking general gratitude and general religiousness into account, so we included a range of wellbeing indicators, including spiritual wellbeing, positive states of mind, and distress.

### 2.1. Method

2.1.1. Participants and Procedure

Participants (N = 267) were recruited from Prolific, an online platform for obtaining research participants. Two attention checks were used during the study, and responses were removed from the final data if they failed either check. Of the 267 participants who took part in the study, 5% (*n* = 14) failed one check and were removed from the data. Therefore, the current analyses are based on 243 participants. These participants ranged from age 18–67 years (M = 35.13, SD = 8.35). The sample skewed toward White individuals (N = 171), but also included 45 Black participants, 12 Multiracial, 8 Asian, 1 Native American, 1 Hawaiian/Other Pacific Islander, and 1 who chose not to say. Seventeen participants identified as Latinx. The sample was predominantly male (N = 151) with 88 female participants. The most popular religion was Roman Catholic (N = 127); others included 39 Non-denominational Christian, 22 Protestant, 21 Agnostic, 19 Other, 3 Muslim, 2 Hindu, 2 Buddhist, 2 Jewish, 1 Mormon, and 1 who chose not to say.

Participants were selected through Prolific under the requirements of being 18 years old or older, fluent in English, living in the U.S., and believing in a deity-based religion

or not being atheist. For that latter requirement, participants were screened for belief in God using the certainty of belief in God item from Rohrbaugh and Jessor (1975). Those who were certain in their belief in the non-existence of God were not eligible to participate. Participants who believed or who had uncertainty about God's existence were included, given that many studies have found that those who may even slightly believe that God exists often report having many feelings related to God (Exline and Rose 2013). Prolific is a newer alternative to Amazon Mechanical Turk (MTurk), which has been shown to be effective for rapid recruitment (Turner et al. 2020) with demonstrated high-quality data (Peer et al. 2022) and more diverse participants than MTurk (Peer et al. 2017). Participants completed the survey in Qualtrics. The amount of missing data was low, with fewer than 2% of items missing responses.

The study received ethical approval before any surveys were released. Respondents were compensated for their time with $5.00 for completing the survey based on an average estimated completion time of 25 min in Qualtrics.

### 2.1.2. Measures

*General gratitude.* General gratitude was assessed with the Gratitude Questionnaire (GQ-6; McCullough et al. 2002). Participants rated each of six items regarding how much they agree or disagree (1—strongly agree, 2—agree, 3—somewhat agree, 4—neither agree nor disagree, 5—somewhat disagree, 6—disagree, 7—strongly disagree). Sample statements include, "I have so much in my life to be thankful for", "If I had to list everything that I felt grateful for, it would be a very long list", and "When I look at the world, I don't see much to be grateful for" (reverse scored). Cronbach's alpha in the present sample was 0.66.

*General religiousness.* The Religious Commitment Inventory-10 (Worthington et al. 2003) was used to assess general religiousness. Participants rated how true of themselves each of 10 statements were, using a 5-point Likert scale (1—not at all true of me to 5—totally true of me). The measure included statements such as: "I often read books and magazines about my faith", "I make financial contributions to my religious organization", and "I spend time trying to grow in understanding of my faith". Cronbach's alpha in the present sample was 0.96.

*Gratitude to God.* GTG was measured using the Religious Gratitude Scale (Krause and Hayward 2015) in which participants were asked to rate on a 4-point scale (1—strongly disagree, 2—disagree, 3—agree, 4—strongly agree) how much they agreed or disagreed with the 4 statements given, including: "I am grateful for God for all He has done for me" and "If I were to make a list of all the things God has done for me, it would be a very long list". Cronbach's alpha in the present sample was 0.96.

*Major life events.* To measure adverse and positive life events, items were aggregated from four life event scales (Lee et al. 2016; Seidlitz and Diener 1993; Berntsen et al. 2011; Brugha and Cragg 1990). Participants indicated whether a positive or negative event had happened to them in their lifetime, scored from never (0), once (1), two times (2), three times (3), four times (4), or five or more times (5). Sample negative events included: "natural disaster", "assault by a stranger", and "domestic violence", while sample positive events included "marriage" and "retirement". Sums were calculated from each list such that higher numbers represented more of each type of event.

*God Images.* Using the Limitless, Authoritarian, Mystical, Benevolent, and Ineffable (LAMBI) scale (Johnson et al. 2019), participants rated on a 1–7 scale (1—strongly disagree, 2—disagree, 3—somewhat disagree, 4—neither agree nor disagree, 5—somewhat agree, 6—agree, 7—strongly agree) to what degree they agreed or disagreed with words that pertain to the following dimensions of God: limitless, authoritarian, mystical, benevolent, ineffable, and no God. Examples of such words included: "Limitless", "Vast", "Boundless", "Wrathful", and "Punishing". Cronbach's alpha in the present study was 0.87 for limitless image, 0.84 for authoritarian image, 0.74 for mystical image, 0.89 for benevolent image, 0.77 for ineffable image, and 0.70 for no God image.

*Religious strain and comfort.* The Attitudes toward God Scale-9 (ATGS-9; Wood et al. 2010) has two subscales, Anger toward God and Comfort from God. Participants rated the

extent to which they agreed with each of nine statements from 1 (not at all) to 5 (extremely). Participants responded to the question: "to what extent do you currently...": regarding items such as "trust God to protect and care for you" (comfort) and "feel angry at God" (anger). Cronbach's alpha in the present sample was 0.93 for anger and 0.97 for comfort.

*Spiritual Wellbeing.* Spiritual wellbeing was measured by the Functional Assessment of Chronic Illness Therapy—Spiritual Wellbeing Scale (FACIT-Sp; Peterman et al. 2002). Participants rated the extent to which they agreed with statements about how their spirituality and faith contributed to their quality of life in the previous seven days using a 0–4 scale ranging from "Not at all" to "Very much". Examples included "I feel peaceful" and "I find comfort in my faith or spiritual beliefs". Cronbach's alpha in the present sample was 0.91.

*Positive States of Mind.* Positive states of mind were measured by the Positive States of Mind scale (PSOM; Horowitz et al. 1988). Participants rated the extent to which they were able to experience the following positive states of mind in the past week: focused attention, productivity, responsible caretaking, restful repose, sensuous nonsexual pleasure, sensuous sexual pleasure, and sharing. Definitions were provided for each state. The measure had participants indicate whether they related to each state of mind as "Unable to experience this even though I wanted to" (1), "Difficult to experience" (2), "Able to experience with only a little difficulty" (3), "Easy to experience" (4) or "Not relevant—have not wanted to experience". Scores for items where participants entered "Not relevant—have not wanted to experience" were omitted from the mean item score for the PSOM. Cronbach's alpha in this sample was 0.80.

*Psychological Distress.* An overall distress score was assessed with the Depression, Anxiety, and Stress Scales-21 (DASS-21; Henry and Crawford 2005). The DASS-21 is a validated 21-item measure of depression, anxiety, and stress that produces an overall measure of psychological distress. The measure had participants rate how much each statement applied to them using a 4-point scale ranging from 0 "Did not apply to me at all" to 3 "Applied to me very much or most of the time". Cronbach's alpha in the present sample was 0.96.

### 2.2. Results

#### 2.2.1. Descriptive Statistics

Table 1 shows descriptive statistics. Some results warrant highlighting (references to scores being high, low, and moderate are made in reference to scale midpoints). Trait GTG and Comfort from God were relatively high. Both adverse and positive events were common, with the typical participant reporting experiencing > 10 each of adverse and positive events over their lifetime. Though general religiousness (RCI-10) was only moderately high, levels of other variables reflecting close relationships with God and positive views of God were quite high, and anger towards God was low.

**Table 1.** Descriptive Statistics for Measures.

| Variable | Scale Range | *M* | *SD* | *Skew* | *Kurtosis* |
|---|---|---|---|---|---|
| GTG | 1 to 7 | 5.84 | 1.43 | −1.65 | 2.31 |
| Adverse Events | 0 to 60 | 15.44 | 11.19 | 1.19 | 1.17 |
| Positive Events | 0 to 45 | 13.79 | 7.08 | 1.90 | 3.90 |
| General Religiousness | 0 to 4 | 2.50 | 1.19 | −0.75 | −0.64 |
| General Gratitude | 1 to 7 | 5.51 | 0.89 | −0.02 | −0.58 |
| God Image: Limitless | 1 to 7 | 5.72 | 1.38 | −1.74 | 3.21 |
| God Image: Authoritarian | 1 to 7 | 4.05 | 1.47 | −0.07 | −0.75 |
| God Image: Mystical | 1 to 7 | 5.61 | 1.01 | −1.02 | 1.88 |
| God Image: Benevolent | 1 to 7 | 5.83 | 1.21 | −1.45 | 2.40 |
| God Image: Ineffable | 1 to 7 | 4.01 | 1.55 | −0.12 | −0.79 |
| God Image: No God | 1 to 7 | 2.69 | 1.41 | 0.54 | −0.35 |
| Comforted by God | 1 to 10 | 8.11 | 2.59 | −1.69 | 1.75 |
| Anger toward God | 1 to 10 | 2.81 | 2.43 | 1.33 | 0.58 |
| Spiritual Wellbeing | 0 to 4 | 2.93 | 0.81 | −0.87 | 0.40 |
| Positive States of Mind | 1 to 7 | 3.42 | 0.52 | −0.91 | 0.34 |
| Psychological Distress | 0 to 3 | 0.82 | 0.67 | 0.65 | −0.57 |

2.2.2. Bivariate Correlations among General Religiousness, General Gratitude, and GTG

Tests of bivariate associations among the three primary predictors in the study indicated that general gratitude and religiousness were positively correlated ($r = 0.72$, $p < 0.001$) and GTG was positively correlated with both general gratitude ($r = 0.52$, $p < 0.001$) and general religiousness ($r = 0.71$, $p < 0.001$).

2.2.3. Partial Correlations Controlling for Demographics (Gender, Age)

To determine the extent to which general gratitude, general religiousness, and GTG differentially related to our set of positive/negative events, God image variables, feelings of anger/comfort toward God, spiritual wellbeing, positive states of mind, and psychological distress, we conducted a series of partial correlations (see Table 2) that controlled for demographics. We conducted Steiger tests (Steiger 1980) to determine whether the magnitudes of correlations differed across general gratitude, general religiousness, and GTG. GTG had a stronger positive correlation with adverse events, positive events, authoritarian image of God, no image of God, anger toward God, and positive states of mind than did general gratitude. GTG had a more positive correlation than did either general religiousness or general gratitude with spiritual wellbeing, comfort from God, and a benevolent image of God. GTG had a stronger negative correlation with ineffable image of God than did general gratitude and a stronger negative correlation with adverse events, positive events, spiritual wellbeing, anger toward God, ineffable image of God, and no image of God than did general religiousness. Interestingly, anger to God was even more strongly negatively correlated with general gratitude than with GTG.

**Table 2.** Partial Correlations Examining whether GTG Shows Distinctive Patterns of Associations Compared to General Gratitude and General Religiousness.

|  | Religiousness | GTG | Gratitude |
|---|---|---|---|
| Adverse Events | −0.04 a | −0.14 *b | −0.24 ***b |
| Positive Events | 0.23 ***a | 0.08 b | −0.13 *c |
| God Image: Limitless | 0.18 **a | 0.34 ***b | 0.33 ***b |
| God Image: Authoritarian | 0.01 a | −0.05 a | −0.24 ***b |
| God Image: Mystical | 0.2 **a | 0.20 **a | 0.16 *a |
| God Image: Benevolent | 0.57 ***a | 0.76 ***b | 0.34 ***c |
| God Image: Ineffable | −0.23 ***a | −0.34 ***b | −0.08 a |
| God Image: No God | −0.27 ***a | −0.46 ***b | −0.46 ***b |
| Comfort from God | 0.74 ***a | 0.9 ***b | 0.22 ***c |
| Anger at God | −0.18 **a | −0.30 ***b | −0.52 ***c |
| Spiritual Wellbeing | 0.65 ***a | 0.74 ***b | 0.43 ***c |
| Positive States of Mind | 0.44 ***a | 0.38 ***a | 0.27 ***b |
| Distress | −0.22 ***a | −0.30 ***a | −0.34 ***a |

Note: letters within rows indicate equivalent correlations based on Steiger tests. Numbers are unstandardized $b$ coefficients. * $p < 0.05$; ** $p < 0.01$; *** $p < 0.001$.

We were interested in examining whether GTG could be distinguished from general religiousness and general gratitude as a predictor of three outcomes (spiritual wellbeing, positive states of mind, and psychological distress). We also tested the interaction effect of general religiousness and GTG with these three outcomes.

2.2.4. Predicting Spiritual Wellbeing

First, we tested the three variables as predictors of spiritual wellbeing. Zero-order correlations revealed similar, large positive correlations of spiritual wellbeing with general religiousness ($r = 0.68$, $p < 0.01$) and GTG ($r = 0.76$, $p < 0.01$) and a medium positive correlation between spiritual wellbeing and general gratitude ($r = 0.33$, $p < 0.01$). We then conducted regression models predicting spiritual wellbeing separately from combinations of gender, age, general religiousness, general gratitude, and GTG. Model 1 included both demographic variables. Model 2a added general religiousness and GTG while Model 2b included demographics, general gratitude, and GTG. Model 3 included all predictors.

Model 4 included all predictors along with the interaction of religiousness and GTG. Results are shown in Table 3.

**Table 3.** Results from Regressions Predicting Spiritual Wellbeing, Positive States of Mind, and Psychological Distress.

| Models Predicting Spiritual Wellbeing | 1 | 2a | 2b | 3 | 4 |
|---|---|---|---|---|---|
| **Predictor** | | | | | |
| Man (vs. Woman) | 0.21 | −0.07 | 0.06 | −0.01 | −0.01 |
| Age | 0.01 | −0.01 * | −0.00 | −0.01 | −0.01 |
| General religiousness | | 0.19 *** | | 0.23 *** | 0.00 |
| General gratitude | | | 0.19 *** | 0.23 *** | 0.21 *** |
| GTG | | 0.32 *** | 0.39 *** | 0.25 *** | 0.21 *** |
| GTG × Religiousness | | | | | 0.04 |
| **Models Predicting Positive States of Mind** | **1** | **2a** | **2b** | **3** | **4** |
| **Predictor** | | | | | |
| Man (vs. Woman) | 0.18 * | 0.05 | 0.14 * | 0.08 | 0.08 |
| Age | 0.01 ** | 0.00 | 0.01 * | 0.01 | 0.01 |
| General religiousness | | 0.16 *** | | 0.18 *** | 0.19 |
| General gratitude | | | 0.09 * | 0.12 ** | 0.12 *** |
| GTG | | 0.05 | 0.12 *** | 0.01 | 0.01 |
| GTG × Religiousness | | | | | −0.00 |
| **Models Predicting Distress** | **1** | **2a** | **2b** | **3** | **4** |
| **Predictor** | | | | | |
| Man (vs. Woman) | 0.16 | 0.24 ** | 0.17 | 0.18 * | 0.18 * |
| Age | 0.01 | 0.01 * | 0.01 | 0.01 | 0.01 |
| General religiousness | | −0.01 | | −0.06 | 0.60 ** |
| General gratitude | | | −0.20 *** | −0.21 *** | −0.17 *** |
| GTG | | −0.14 ** | −0.10 *** | −0.07 | 0.04 |
| GTG × Religiousness | | | | | −0.11 *** |

Note. Numbers are unstandardized *b* coefficients. * $p < 0.05$; ** $p < 0.01$; *** $p < 0.001$.

### 2.2.5. Predicting Positive States of Mind

Next, we tested the three variables as predictors of positive states of mind. Zero-order correlations revealed similarly modest, positive correlations between positive states of mind and religiousness and GTG, with a small, positive correlation between general gratitude and positive states of mind: religiousness ($r = 0.50$, $p < 0.01$), GTG ($r = 0.43$, $p < 0.01$), and general gratitude ($r = 0.21$, $p < 0.01$). We followed the same sequence of models that we used with spiritual wellbeing, described above. Results are shown in Table 3.

### 2.2.6. Predicting Psychological Distress

Finally, we tested the three variables as predictors of psychological distress. Zero-order correlations revealed similarly modest, negative correlations between psychological distress and general gratitude and GTG, with a small, negative correlation between religiousness and psychological distress: general gratitude ($r = −0.36$, $p < 0.01$), GTG ($r = −0.25$, $p < 0.01$), and religiousness ($r = −0.15$, ns). We followed the same sequence of models that we used with spiritual wellbeing and positive states of mind (see Table 3) and found a significant interaction effect where general religiousness moderated the relationship between GTG and psychological distress such that high religiousness predicted less distress in individuals higher in GTG and more distress in individuals lower in GTG. For individuals low in religiousness, distress was less associated with GTG. The interaction is plotted in Figure 1.

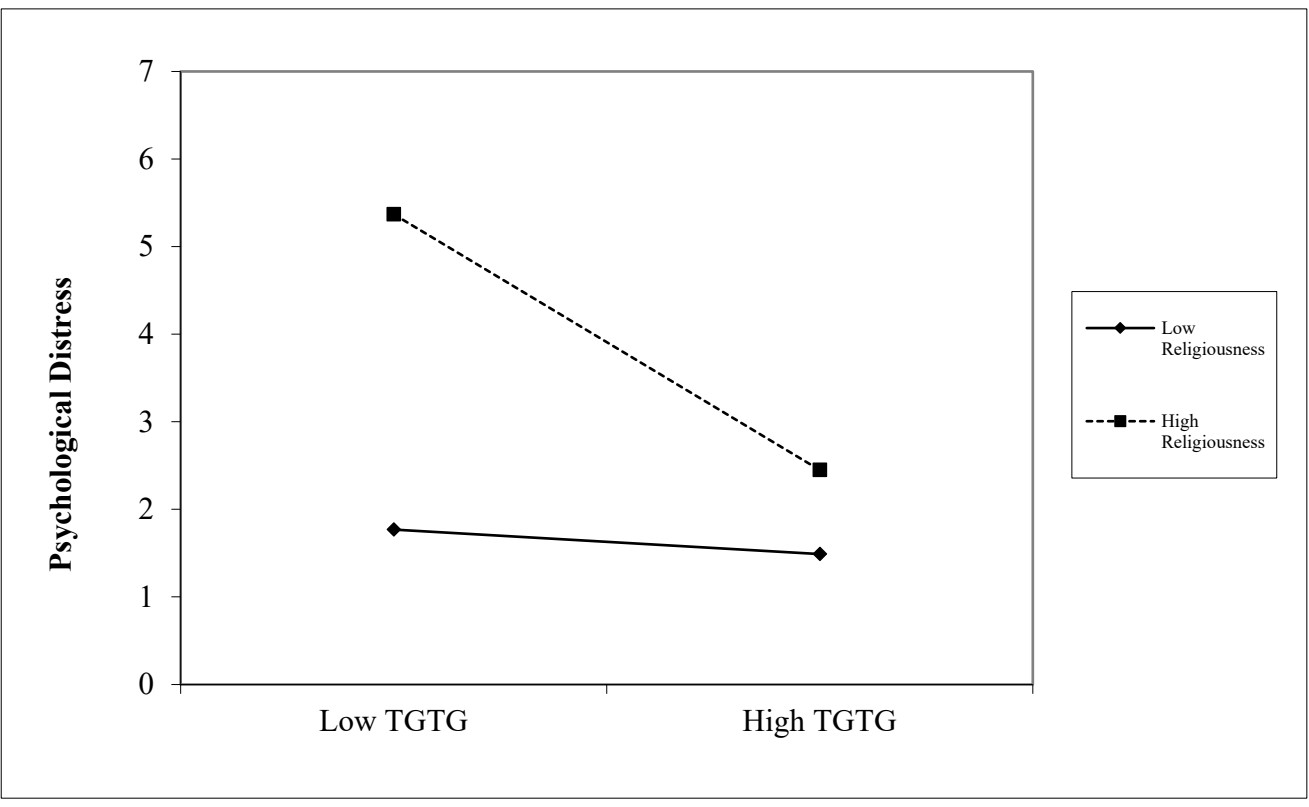

**Figure 1.** Interaction between General Religiousness and Gratitude to God Predicting Levels of Psychological Distress.

*2.3. Discussion*

Our aims in this study were to use two different methods for determining the extent to which GTG constituted a unique construct. First, we examined how associations of GTG with life events, religious variables, and outcomes differed from those of general religiousness and general gratitude, and then we examined how GTG predicted outcomes when taking into account general religiousness and general gratitude.

The correlational results suggest that GTG is indeed a unique construct: The associations of GTG with the targeted constructs were in some cases similar to general religiousness, and in other cases, with general gratitude, but in many respects the associations of GTG were also quite different from those of general religiousness or general gratitude. For example, more lifetime negative events were unrelated to religiousness but related to less GTG and even more so to less general gratitude, while experiencing more positive lifetime events was positively associated with general religiousness, unrelated to GTG, and inversely related to general gratitude. GTG was substantially more strongly related to benevolent aspects of religiousness such as benevolent God images and comfort from God.

Similarly, the results of the second approach—differential prediction of outcomes by GTG, general religiousness, and general gratitude—also suggest that these constructs are distinct. In predicting spiritual wellbeing, all three constructs contributed substantial variance, suggesting that the effects of GTG were not subsumed by general religiousness or general gratitude. On the contrary, GTG did not contribute unique variance to positive states of mind, but in predicting psychological distress, both GTG and general gratitude predicted unique variance in lower distress. Further, when all three constructs were entered in the model, general religiousness actually predicted *higher* distress. This latter finding might suggest that religiousness without gratitude is a product of or leads to more distress. In fact, we found the association between psychological distress and GTG was even stronger when taking general religiousness into account—those high on both GTG *and* religiousness

had much less psychological distress while for those with low general religiousness, GTG did not appear related to psychological distress.

These results must be interpreted within the limitations of the study. The study used a cross-sectional design, precluding examination of temporal ordering. GTG may, for example, lead to subsequent lower levels of psychological distress and more positive states of mind, or less distress and more positive states of mind may lead to more subsequent GTG. In addition, the measures of adversity referred to individuals' whole lives; it is likely that substantial retrospective bias influenced recollection of lifetime events (Belli 1998). In addition, Study 1 had limited ecological validity related to psychological distress, a weakness we addressed in Study 2. In spite of these limitations, the promising results of Study 1 suggest GTG is a unique construct, distinct from general religiousness and general gratitude, warranting future research.

## 3. Study 2

In Study 2, we aimed to replicate and build on the findings from Study 1. The first aim was to determine if GTG differentially related to a variety of constructs that might be especially relevant to it compared to general gratitude or general religiousness. We again included life event history, particularly major positive and negative events, but measured them more proximally to limit retrospective recall bias, again expecting that people who have experienced more recent good fortune (i.e., more positive life events) and less adversity (i.e., fewer negative life events) would feel more gratitude. We again included experiential aspects of religiousness, such as images of God and finding comfort from or feeling anger towards God. We also included a measure of locus of control, given that some theory has linked locus of control to general feelings of gratitude (Watkins et al. 2003), positing that those with an internal sense of control may be less likely to experience gratitude as they may see benefits more as a result of their own effort. However, the few studies that have examined this issue demonstrated that an internal locus of control predicts *more* general gratitude (Kashdan et al. 2009; Watkins et al. 2003); further, in Watkins and colleagues' study of college students, *divine* locus of control was even more strongly associated with general gratitude than was internal locus of control (Watkins et al. 2003). These intriguing findings suggest that individuals with an internal locus of control may experience that control as secondary (i.e., in conjunction with God's control). However, very little research is available on this topic, none specific to GTG.

As our second aim, we built on Study 1 findings by examining whether GTG was associated with wellbeing. However, because Study 1 collected all data cross-sectionally, the temporal association between GTG and wellbeing could not be determined. In Study 2, we examined associations between GTG and *subsequent* wellbeing after taking general gratitude and general religiousness into account, and rather than examining general wellbeing, we looked at more proximal short-term wellbeing aggregated across daily assessments over a two-week period.

### 3.1. Method

3.1.1. Participants and Procedure

Participants consisted of 184 individuals (51.35% female, 44.32% male, 1.08% transgender male, 2.7% non-binary/third gender) aged 18–74 (m = 33.8) who reside in the U.S. and are registered to the online survey site Prolific. Participants self-selected to take part in the study. The inclusion criteria were that the participant had to be at least 18 years old, read and write fluently in English, live in the U.S., and believe in a deity-based religion (Pew Research Center). For that latter requirement, participants were screened for belief in God using the certainty of belief in God item from Rohrbaugh and Jessor (1975). The study received ethical approval from the University of Connecticut Institutional Review Board. Respondents were compensated for their time with $4.00 for completing the baseline survey and $1.50 for each of the 14 daily surveys completed. All constructs were assessed at baseline except for daily positive and negative affect.

### 3.1.2. Measures

*Major life events*. The same positive and negative life events items were used as in Study 1, but participants were asked to indicate if each positive or negative event had happened less than three months ago (1), less than 6 months ago (2), or had not happened (3). The 42 events in the measure included: "natural disaster", "assault by a stranger", "domestic violence", "marriage", "retirement", "took a vacation", and "major financial crisis". We combined categories 1 and 2 into a single category and scored each event dichotomously: 0 = did not happen, 1 = happened within 6 months. We then created a total score for adverse events (possible range = 0 to 26) and positive events (possible range = 0 to 15).

*Locus of control*. We used both the Locus of Control Scale (LoC; Lumpkin 1985), which measures internal/external locus of control and the God as a Causal Agent Scale (GCAS; Ritzema and Young 1983), which measures explicitly divine control. In the LoC scale, participants rated the extent to which they agreed with 6 statements such as: "When I make plans, I am almost certain that I can make them work" and "What happens to me is my own doing" on a 5-point Likert scale (1—strongly disagree, 2—somewhat agree, 3—neither agree nor disagree, 4—somewhat disagree, 5—strongly disagree). On the GCA Scale, participants rated the extent to which they agreed with 14 statements such as "God Created the world by giving the commands", "Miracles happen much more frequently than most people suspect", and "I'm usually skeptical when someone tells me that they're convinced that God did something to change their attitudes or beliefs" on a 5-point Likert scale (1—agree, 2—somewhat agree, 3—neutral, 4—somewhat disagree, 5—disagree).

*General Religiousness*. This construct was assessed with the same measure as in Study 1, the Religious Commitment Inventory-10 (Worthington et al. 2003).

*Anger towards and Comfort from God.* These constructs were measured with the same ATGS-9 (Wood et al. 2010) as in Study 1.

*God Image.* The same scale (LAMBI Johnson et al. 2019) used in Study 1 was also used in Study 2.

*Gratitude to God.* GTG was measured with the same scale used in study 1, the Religious Gratitude Scale (Krause and Hayward 2015).

*General Gratitude*. General gratitude was assessed with the same measure as in Study 1, the GQ-6 (McCullough et al. 2002).

*Daily Positive/Negative Affect.* Each daily survey contained a measure of affect using the SPANE (Diener et al. 2009). Participants were asked to rank on a 5-point scale (1—very slightly or not at all, 2—slightly, 3—somewhat, 4—moderately, 5—extremely) how much they were experiencing 12 feelings today including: "Positive", "Negative", "Good", "Bad", "Pleasant", and "Unpleasant"). We aggregated across days to compute average levels of daily positive/negative affect over two weeks.

### 3.1.3. Procedures

The baseline took approximately 35 min to complete and each daily survey took a few minutes.

### *3.2. Results*

### 3.2.1. Data Analysis Plan

The main data analysis plan was similar to that of Study 1: We computed descriptive statistics, examined patterns of correlations between GTG, general gratitude, and religiousness with various constructs, and then examined whether GTG predicted relevant outcomes beyond general gratitude and general religiousness. Prior to the main analyses, because locus of control measures showed poor psychometric characteristics in previous research, we conducted data reduction analyses to help create scales with acceptable levels of internal consistency.

### 3.2.2. Data Reduction for Locus of Control Scales

Although the LoC scale was designed to be a unitary measure, previous research showed poor psychometric characteristics of a composite scale (Lange and Tiggemann 1981). We therefore explored the structure of the items in the current study. Examination of scree plots and parallel analysis (Horn 1965) suggested that two factors were present in the data. Exploratory factor analysis (EFA) using the minimal residual method and direct oblimin rotation showed that the first factor consisted of the three items where higher scores indicated greater internal locus of control, and the second factor contained the three items where higher scores indicated greater external locus of control. We used items from these two factors as separate measures in this study.

Similarly, because previous work showed that composite measures of items from the GCAS (Ritzema and Young 1983) exhibited poor psychometric characteristics (Jackson and Coursey 1988), we conducted exploratory analyses of the items. Examination of scree plots and parallel analysis (Horn 1965) suggested that one or two factor solutions were acceptable. The one-factor EFA resulted in several items with low factor loadings (<0.30). In the two-factor EFA, we selected the seven items with strong loadings on the first factor (>0.30) to create our measure of God locus of control. We did not create a measure for items on the second factor due to the lack of theoretical rationale and empirical evidence for multifactorial solutions of God locus of control.

### 3.2.3. Descriptive Statistics

Table 4 shows descriptive statistics. Some results warrant highlighting (references to scores being high, low, and moderate are made with regard to scale midpoints). Both GTG and general gratitude levels were relatively high. Recent individual adverse and positive events were rare; the typical participant reported experiencing just over one each of both positive and negative life events. Levels of all aspects of locus of control were moderate to high. Participants' levels of general religiousness were moderate, but variables reflecting close relationships with God and positive views of God were quite high while anger toward God was low.

**Table 4.** Descriptive Statistics for Baseline Measures and Aggregated Positive/Negative Affect.

| Variable | Scale Range | *M* | *SD* | *Skew* | *Kurtosis* | *α* |
|---|---|---|---|---|---|---|
| GTG | 1 to 4 | 3.35 | 0.72 | −1.16 | 0.96 | 0.95 |
| Adverse Events | 0 to 26 | 1.43 | 1.93 | 2.59 | 8.72 | 0.68 |
| Positive Events | 0 to 15 | 1.20 | 1.32 | 1.41 | 2.26 | 0.45 |
| External LoC | 0 to 4 | 3.27 | 0.80 | −0.02 | −0.39 | 0.53 |
| Internal LoC | 0 to 4 | 3.40 | 0.77 | −0.28 | −0.15 | 0.60 |
| God LoC | 0 to 5 | 3.87 | 0.97 | −0.80 | −0.12 | 0.91 |
| General Religiousness | 0 to 4 | 1.78 | 1.09 | 0.23 | −1.05 | 0.94 |
| General Gratitude | 1 to 7 | 5.71 | 1.00 | −0.70 | 0.26 | 0.79 |
| God Image: Limitless | 1 to 7 | 6.21 | 1.12 | −1.95 | 4.57 | 0.93 |
| God Image: Authoritarian | 1 to 7 | 3.89 | 1.52 | −0.08 | −0.67 | 0.87 |
| God Image: Mystical | 1 to 7 | 5.73 | 1.24 | −1.42 | 2.62 | 0.88 |
| God Image: Benevolent | 1 to 7 | 6.26 | 1.08 | −2.28 | 6.54 | 0.93 |
| God Image: Ineffable | 1 to 7 | 4.15 | 1.69 | −0.14 | −0.92 | 0.95 |
| God Image: No God | 1 to 7 | 3.85 | 1.35 | 2.03 | 4.08 | 0.95 |
| Comforted by God | 1 to 5 | 3.96 | 1.11 | −1.07 | 0.34 | 0.95 |
| Anger toward God | 1 to 5 | 1.65 | 0.87 | 1.38 | 1.08 | 0.89 |
| Aggregate Positive Affect | 1 to 5 | 3.11 | 0.96 | −0.01 | −0.82 | 0.95 |
| Aggregate Negative Affect | 1 to 5 | 1.55 | 0.52 | 2.12 | 7.81 | 0.89 |

### 3.2.4. Bivariate Correlations among General Religiousness, General Gratitude, and GTG

Tests of bivariate associations among the three primary predictors in the study indicated that general gratitude and religiousness were positively correlated ($r = 0.29$, $p < 0.001$)

and GTG was positively correlated with both general gratitude ($r = 0.52$, $p < 0.001$) and general religiousness ($r = 0.43$, $p < 0.001$).

### 3.2.5. Partial Correlations Controlling for Demographics (Gender, Age)

To examine whether GTG relates to other variables differently than general gratitude or religiousness, we conducted a series of partial correlations (controlling for age and gender).

Table 5 shows these associations between general gratitude, general religiousness, and GTG with recent positive and negative events, locus of control variables, God image variables, and feeling anger/comfort toward God. We conducted Steiger tests (Steiger 1980) to determine whether the magnitudes of correlations differed. GTG had higher, positive correlations with God locus of control, LAMBI-benevolent, and comforted by God than did general gratitude or general religiousness, as well as more negative correlations with LAMBI-no God. For other variables, GTG had similar magnitudes of correlations with general gratitude, general religiousness, or both.

**Table 5.** Partial Correlations Examining whether GTG Shows Distinctive Patterns of Associations Compared to General Gratitude and Religiousness.

|  | General Gratitude | General Religiousness | GTG |
|---|---|---|---|
| Positive Experiences | −0.04 a | 0.09 a | 0.10 a |
| Negative Experiences | −0.05 a | 0.25 ***b | 0.09 a |
| Internal LoC | 0.17 *b | −0.15 *a | 0.09 b |
| External LoC | −0.31 ***a | −0.10 b | −0.30 ***a |
| God LoC | 0.43 ***a | 0.47 ***a | 0.78 ***b |
| LAMBI Limitless | 0.35 ***a | 0.37 ***a | 0.58 ***b |
| LAMBI Authoritarian | −0.05 a | 0.21 **b | 0.27 ***b |
| LAMBI Mystical | 0.08 a | −0.08 a | 0.09 a |
| LAMBI Benevolent | 0.32 ***a | 0.30 ***a | 0.60 ***b |
| LAMBI Ineffable | −0.13 a | −0.24 ***ab | −0.34 ***b |
| LAMBI No God | −0.32 ***a | −0.36 ***a | −0.62 ***b |
| Anger Toward God | −0.46 ***a | −0.13 b | −0.26 ***b |
| Comforted by God | 0.50 ***a | 0.43 ***a | 0.82 ***b |

Note: letters within rows indicate equivalent correlations based on Steiger tests. * $p < 0.05$. ** $p < 0.01$. *** $p < 0.001$.

### 3.2.6. Predicting Aggregate Positive and Negative Affect

Next, we examined whether GTG could be distinguished from general religiousness and general gratitude as a predictor of daily affect. Thus, we examined all three variables as predictors of aggregate daily PA and NA. Zero-order correlations revealed similar, positive associations between PA and each predictor: GTG ($r = 0.34$, $p < 0.001$), religiousness ($r = 0.27$, $p < 0.001$), and gratitude in general ($r = 0.28$, $p < 0.001$). NA also had significant zero-order associations with GTG ($r = -0.19$, $p = 0.01$) and general gratitude in ($r = -0.25$, $p < 0.001$) but not with general religiousness ($r = 0.04$, $p = 0.59$).

We then conducted regression models predicting aggregate daily PA and NA separately from combinations of gender, age, baseline religiousness, trait gratitude, and GTG. Model 1 included both demographic variables. Model 2a added religiousness and GTG. Model 2b included demographics, trait gratitude, and GTG. Model 3 included all predictors. Model 4 included all predictors and the interaction of religiousness and GTG. Results are shown in Table 6. For PA, being a man and having higher levels of GTG were robust, positive predictors. For NA, none of the variables were consistent predictors.

**Table 6.** Results from Regressions Predicting Positive Affect and Negative Affect.

| Models Predicting PA | 1 | 2a | 2b | 3 | 4 |
|---|---|---|---|---|---|
| **Predictor** | | | | | |
| Man (vs. Woman) | 0.44 ** | 0.37 ** | 0.40 ** | 0.38 ** | 0.38 ** |
| Age | 0.00 | 0.00 | 0.00 | 0.00 | 0.00 |
| General religiousness | | 0.09 | | 0.09 | 0.06 |
| General gratitude | | | 0.11 | 0.10 | 0.10 |
| GTG | | 0.41 *** | 0.39 ** | 0.33 ** | 0.35 ** |
| Religiousness × GTG | | | | | 0.13 |
| **Models Predicting NA** | | | | | |
| **Predictor** | | | | | |
| Man (vs. Woman) | −0.02 | −0.02 | −0.02 | −0.04 | −0.03 |
| Age | 0.00 | 0.00 | 0.00 | 0.00 | 0.00 |
| General Religiousness | | 0.06 | | 0.06 | 0.07 |
| General Gratitude | | | −0.09 | −0.09 * | −0.09 |
| GTG | | −0.15 * | −0.04 | −0.08 | −0.10 |
| Religiousness × GTG | | | | | −0.05 |

Note. Numbers are unstandardized $b$ coefficients. * $p < 0.05$; ** $p < 0.01$; *** $p < 0.001$.

### 3.3. Discussion

Results of Study 2 were in many ways similar to and added to Study 1 findings. As in Study 1, we found strong evidence that GTG is a unique construct. Comparing associations with a range of different constructs suggests differential patterns for GTG, general religiousness and general gratitude. Once again, we found notably stronger associations of GTG with benevolent aspects of religiousness. Particularly striking was the strong association between GTG and divine locus of control. Future research is needed to understand why these two constructs are so closely related; perhaps divine locus of control leads people to attribute positive occurrences to God and thus they feel more grateful to God.

In terms of predicting subsequent wellbeing in the form of positive and negative affect, results were surprising and quite different from the findings in Study 1, which focused on more general and global aspects of wellbeing. In particular, neither general gratitude nor general religiousness predicted aggregated positive affect over the subsequent two weeks, but GTG was a very strong positive predictor of positive affect. However, neither general religiousness nor GTG related to negative affect, and general gratitude was only a modest (negative) predictor. These findings are inconsistent with previous work demonstrating robust salutary associations of gratitude and wellbeing (Portocarrero et al. 2020). However, we did find salutary associations between gratitude and affect before taking into account the demographics and other variables. When considered together, only GTG predicted positive affect. Unlike in Study 1, effects of GTG on wellbeing did not interact with general religiousness.

These findings should be interpreted within the context of study strengths and limitations. Building on Study 1, we considered a similar set of variables and also included multiple dimensions of locus of control. We were able to use a more ecologically valid method, daily diaries, to look at temporal relations between GTG, general religiousness, and general gratitude with measures of subsequent positive and negative affect. However, the longitudinal component of the study was very short, allowing us only a brief glimpse into the associations among variables. Further, we did not have more fine-grained detail regarding the attributions people make for positive and negative events.

### 4. Overall Discussion

The findings from this set of studies provide converging evidence for the assertion that GTG is a distinct construct and not simply redundant with either general gratitude or general religiousness. This distinctiveness is suggested both by findings from the correlational analyses showing differential strengths of associations between a host of religious and nonreligious variables with general religiousness, general gratitude, and GTG

and by findings showing that GTG predicts aspects of wellbeing in ways that neither general religiousness nor general gratitude do. This distinctiveness is important in establishing GTG as a unique construct worthy of more intensive study, demonstrating that GTG is not simply an aspect of already well-recognized constructs.

In regard to this first set of analyses, considering associations of GTG with a host of other constructs, we found across two studies that GTG appeared to have stronger relations with positive aspects of experiential religiousness such as a loving image of God or finding comfort in God than did either general religiousness or general gratitude. These close associations between viewing God as benevolent and loving and feeling grateful specifically to God may reflect a recursive process in which experiencing GTG is both based on and reinforces warm, positive views of God. The findings across both studies that GTG was less strongly inversely related to anger to God than general gratitude might reflect the potentially complex nexus of feeling grateful specifically to God but also potentially viewing God as ultimately in charge of the world. Anger to God is usually associated cross-sectionally with greater psychological distress but its resolution sometimes leads to greater spirituality or closeness with God (Wilt et al. 2017).

With regard to differential prediction of wellbeing by GTG, the strong and distinct associations of GTG with both positive and negative aspects of wellbeing across the two studies suggest again that GTG may make an important contribution to wellbeing over and above general religiousness and general gratitude rather than simply being redundant with them, which is further support for the notion that GTG comprises an important construct in and of itself.

In addition to shedding light on the unique nature of GTG, many of the specific findings and discrepancies across the two studies are interesting and warrant further inquiry. For example, we found individuals who reported higher lifetime adverse events had lower general gratitude as well as GTG, but those with more recent adverse events reported *more* general gratitude. Positive events were generally unrelated to gratitude but lifetime positive events were related to higher religiousness, yet not to GTG or general gratitude. Studies that delve deeper into these associations, perhaps following people over substantial periods of time, as well as studies inquiring about attributions and implications of positive and negative events, may help illuminate associations between life events and GTG.

The present set of studies were highly exploratory, given the lack of previous research on the topic of the uniqueness of GTG as a construct, and findings must be interpreted within the limitations of the studies. We relied on self-report correlational data, which suffers from method invariance and an inability to make causal inferences. The generalizability of our samples is also limited. Especially when studying constructs like GTG, it is important to consider how results might vary in different groups, such as people from different religious traditions. Future research on these topics should consider longitudinal and experimental research to better understand the directionality and causal nature of associations. Such work may include additional constructs of interest and focus on broader groups, extending this area of work to other ethnicities, cultures, and faith traditions. Based on our results, GTG appears to be a promising construct that warrants additional research attention to illuminate its unique and powerful properties.

**Author Contributions:** Conceptualization, C.L.P.; methodology: C.L.P. and J.A.W.; formal analysis, J.A.W. and A.B.D.; writing—original draft preparation, C.L.P., J.A.W. and A.B.D.; writing—review and editing, C.L.P., J.A.W. and A.B.D.; funding acquisition, C.L.P. All authors have read and agreed to the published version of the manuscript.

**Funding:** John Templeton Foundation Grant 61513.

**Institutional Review Board Statement:** The study was approved by the University of Connecticut IRB (Protocol# X21-0125) on June 17 2021.

**Informed Consent Statement:** Informed consent was obtained from all subjects involved in the study.

**Data Availability Statement:** Data are available upon request from the first author.

**Conflicts of Interest:** The authors declare no conflict of interest.

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
