# Peer review of "Gratitude to God: A Unique Construct Adding to Our Understanding of Religiousness and Gratitude"

_religions, doi:10.3390/rel13090872_

Round 1

Reviewer 1 Report

On p. 2, line 90 it should read "two of Iranian Muslims in Iran" (not in Iraq).

Shouldn’t you exclude Agnostic people from your analysis, like people who were atheist?

Shouldn’t you mention, somewhere, that what the acronym LAMBI stands for?

I would like to see zero-order correlations among your main variables (religiousness, GTG, and Gratitude) somewhere in the Results.

Author Response

On p. 2, line 90 it should read "two of Iranian Muslims in Iran" (not in Iraq).

AU: Thank you for noting this; we have thus corrected it.

Shouldn’t you exclude Agnostic people from your analysis, like people who were atheist?

AU: This is a good question; we now provide information on why we did not exclude atheists.  Essentially, for both studies, participants were screened for belief in God using the certainty of belief in God item from Rohrbaugh and Jessor (1975). Those who are certain in their belief in the non-existence of God were not be eligible to participate. Participants who believe or who have uncertainty about God’s existence were included, given that many studies have found that those who may even slightly believe that God exists often report having many feelings related to God (Exline & Rose, 2013).

Shouldn’t you mention, somewhere, that what the acronym LAMBI stands for?

Thanks for this suggestion; we now use the scale name at first mention.

I would like to see zero-order correlations among your main variables (religiousness, GTG, and Gratitude) somewhere in the Results.

AU: We have added this information for both studies.

Reviewer 2 Report

An excellent research article. The limitations of the study are clearly stated, relevant literature explored, ethical protections provided, and conclusions clearly stated. In the particular issue, this article will contribute much. 

Two concerns— the first I hope will be addressed. After reading the article, the methods, and the conclusions, I appreciated the work, but wondered “so what?”  I know this is exploratory research, yet I am not clear what difference it’s findings make. Gratitude to God and simple gratitude would always seem to be different. So — what are the next steps to which the article directs us, how does it help us in the study of religion or theology make the next steps in our research. A 2 or 3 paragraph section dealing with this at the end and an early paragraph in first page of the article would enhance the interest of readers and help the reader move to next steps.

Second, I think the authors misunderstand anger to God. Yes, sometimes that anger can distance us from God. We’ll can tell stories.  Yet at other times, anger simply shows the intimacy and trust of a person’s connection to God. The connection of anger and intimacy are seen in many texts of the Hebrew Bible, used by Jews and Christians. The authors need to be more nuanced here.

Honestly a fine article and worth publishing, yet attention to the two above concerns would enhance interest and impact. 

Author Response

Two concerns— the first I hope will be addressed. After reading the article, the methods, and the conclusions, I appreciated the work, but wondered “so what?”  I know this is exploratory research, yet I am not clear what difference it’s findings make. Gratitude to God and simple gratitude would always seem to be different. So — what are the next steps to which the article directs us, how does it help us in the study of religion or theology make the next steps in our research. A 2 or 3 paragraph section dealing with this at the end and an early paragraph in first page of the article would enhance the interest of readers and help the reader move to next steps.

AU: We agree that conceptually the distinctiveness of GTG is obvious and thus have endeavored to make the value of identifying the distinctiveness of GTG empirically to promote future study. 

Second, I think the authors misunderstand anger to God. Yes, sometimes that anger can distance us from God. We’ll can tell stories.  Yet at other times, anger simply shows the intimacy and trust of a person’s connection to God. The connection of anger and intimacy are seen in many texts of the Hebrew Bible, used by Jews and Christians. The authors need to be more nuanced here.

AU: Thank you for raising this important point. We agree that anger is not always harmful and may reflect intimacy and trust, etc. We have attempted to provide more nuance in the manuscript.